# Artificial Intelligence in the Diagnosis of Oral Diseases: Applications and Pitfalls

**DOI:** 10.3390/diagnostics12051029

**Published:** 2022-04-19

**Authors:** Shankargouda Patil, Sarah Albogami, Jagadish Hosmani, Sheetal Mujoo, Mona Awad Kamil, Manawar Ahmad Mansour, Hina Naim Abdul, Shilpa Bhandi, Shiek S. S. J. Ahmed

**Affiliations:** 1Department of Maxillofacial Surgery and Diagnostic Sciences, Division of Oral Pathology, College of Dentistry, Jazan University, Jazan 45142, Saudi Arabia; 2Department of Biotechnology, College of Science, Taif University, Taif 21944, Saudi Arabia; dr.sarah@tu.edu.sa; 3Department of Diagnostic Dental Sciences, Oral Pathology Division, Faculty of Dentistry, College of Dentistry, King Khalid University, Abha 61411, Saudi Arabia; jhosmani@kku.edu.sa; 4Division of Oral Medicine & Radiology, College of Dentistry, Jazan University, Jazan 45142, Saudi Arabia; sheetalmujoo@yahoo.co.uk; 5Department of Preventive Dental Science, College of Dentistry, Jazan University, Jazan 45142, Saudi Arabia; munakamil@yahoo.com; 6Department of Prosthetic Dental Sciences, College of Dentistry, Jazan University, Jazan 45142, Saudi Arabia; ahmad955mls@gmail.com (M.A.M.); drhinaprostho@gmail.com (H.N.A.); 7Department of Restorative Dental Sciences, Division of Operative Dentistry, College of Dentistry, Jazan University, Jazan 45142, Saudi Arabia; shilpa.bhandi@gmail.com; 8Multi-Omics and Drug Discovery Lab, Chettinad Academy of Research and Education, Chennai 600130, India; shiekssjahmed@gmail.com

**Keywords:** artificial intelligence, artificial neural network, diagnosis, deep learning, machine learning, oral diseases

## Abstract

*Background:* Machine learning (ML) is a key component of artificial intelligence (AI). The terms machine learning, artificial intelligence, and deep learning are erroneously used interchangeably as they appear as monolithic nebulous entities. This technology offers immense possibilities and opportunities to advance diagnostics in the field of medicine and dentistry. This necessitates a deep understanding of AI and its essential components, such as machine learning (ML), artificial neural networks (ANN), and deep learning (DP). *Aim:* This review aims to enlighten clinicians regarding AI and its applications in the diagnosis of oral diseases, along with the prospects and challenges involved. Review results: AI has been used in the diagnosis of various oral diseases, such as dental caries, maxillary sinus diseases, periodontal diseases, salivary gland diseases, TMJ disorders, and oral cancer through clinical data and diagnostic images. Larger data sets would enable AI to predict the occurrence of precancerous conditions. They can aid in population-wide surveillance and decide on referrals to specialists. AI can efficiently detect microfeatures beyond the human eye and augment its predictive power in critical diagnosis. *Conclusion:* Although studies have recognized the benefit of AI, the use of artificial intelligence and machine learning has not been integrated into routine dentistry. AI is still in the research phase. The coming decade will see immense changes in diagnosis and healthcare built on the back of this research. *Clinical significance:* This paper reviews the various applications of AI in dentistry and illuminates the shortcomings faced while dealing with AI research and suggests ways to tackle them. Overcoming these pitfalls will aid in integrating AI seamlessly into dentistry.

## 1. Introduction

Artificial intelligence (AI) and machine learning (ML) are terms that are often used in research that are interchangeable even though they have different meanings. John McCarthy, called the father of artificial intelligence, coined the term ‘artificial intelligence’ to describe machines with the potential to perform actions that were considered intelligent without any human intervention [1]. These machines are capable of solving problems based on the data input. Artificial intelligence has long been the mainstay of popular science fiction. It originally stemmed from Alan Turing’s “Imitation game” or the “Turing test” [2]. Logic Theorist, developed by Allen Newell and Herbert Simon in the year 1955, was the first-ever AI program [3].

Machine learning (ML) is a subset of artificial intelligence [4]. Simon Cowell coined the term in 1959 [5]. ML predicts the outcome based on the dataset provided to it using algorithms, such as artificial neural networks (ANN). These networks mimic the human brain and have interconnected artificial neurons that receive and analyze data signals. Warren McCulloch and Walter Pitts suggested this concept in a seminal paper published in 1943. Later, Minsky and Dean Edmunds developed the first ANN, the stochastic neural analog reinforcement calculator, in 1951 [6].

Convolutional neural network (CNN) or deep learning (DL) is an approach in ML introduced in 2006 by Hinton et al. [7]. It utilizes multi-layer neural networks to compute data. Deep learning algorithms have the potential to analyze patterns based on the data and improve the outcome. The development of the backpropagation algorithm in 1969 paved the way for deep learning systems [8]. Figure 1 depicts the important milestones in the advancement of AI through the years.

An abundant supply of data sets is crucial for implementing machine learning. Data can refer to a variety of inputs: it can be images in the form of clinical photographs, radiographs, text in the form of patient data, patient symptoms information, and audio in the form of voice, murmurs, bruits, auscultation, or percussion sound. Figure 2 shows the working of AI in a schematic format. Adaptability to a variety of inputs in artificial intelligence added advantage to revolutionizing medical, dental, and healthcare delivery. Recently, artificial intelligence in dentistry alone has created immense attention in specialties such as orthodontics [9,10,11], endodontics [12,13], prosthodontics [14,15], restorative dentistry [16,17], periodontics [18,19,20], oral and maxillofacial surgery [21,22,23]. Research reveals promising results, although most applications are in the developmental phase. It becomes a necessity that dentists need to understand the foundational concepts and applications of AI in dentistry to adapt to a changing healthcare landscape [24].

Today, artificial intelligence (AI) has been suggested useful in disease diagnosis, predicting prognosis, or developing patient-specific treatment strategies [25]. Particularly, AI can assist dentists in making time-sensitive critical decisions. It can remove the human element of error in decision-making, providing a superior and uniform quality of health care while reducing the stress load on the dentists. This paper reviews the available literature selected that are pertaining to the research and development of AI in the diagnosis of various oral and maxillofacial diseases, such as dental caries, periodontal disease, maxillary sinus diseases, salivary gland diseases, temporomandibular joint disorders, osteoporosis, and oral cancer.

## 2. Search Strategy

We used PubMed, Google Scholar, and ScienceDirect to conduct a systematic search using a variety of key terms that included “Convolutional Neural Network” or “Deep Learning” or “Natural Language Processing” OR “neural network” OR “Machine Learning” OR “unsupervised learning” OR “Artificial Intelligence” OR “supervised learning” for the model. Similarly, for disease, the terms included “dental caries” OR “periodontal disease” OR “maxillary sinus diseases” OR “salivary gland diseases” OR “Temporomandibular joint disorders” OR “osteoporosis” OR oral cancer. We looked for articles that were published between January 2016 and December 2021. In addition to the search, the reference lists of the selected article were examined to add an article for this review.

## 3. Dental Caries

Dental caries is the most prevalent disease across the globe. Early diagnosis is key in decreasing caries-related indisposition in patients. Caries diagnosis is exceedingly based on visual cues and radiographic data. This visual data can be a form of input dataset for machine learning (ML). Devito et al. (2008) evaluated the efficiency of a multi-layer perceptron neural network in diagnosing proximal caries in bitewing radiographs and concluded that the diagnostic improvement was 39.4% [26]. Lee et al. (2018) used 3000 periapical radiographs to evaluate the efficacy of deep convolutional neural networks to identify dental caries. High accuracy of 89%, 88%, and 82% was observed in the premolar, molar, and both the premolar-molar regions [27]. Hung et al. (2019) conducted a study with the test and training set comprised of data obtained from the National Health and Nutrition Examination Survey. Supervised learning methods were used to classify the data based on the presence or absence of root caries. Among the various ML methods used in their study, the support vector machine (SVM) showed the best performance in identifying root caries [28].

Similarly, the clinical imaging data from various sources have been used in AI models for diagnosing dental caries with excellent results. In 2019, a study examined the use of convolutional neural networks (CNN) to identify dental caries in near-infrared transillumination images. CNN increased the speed and accuracy of caries detection [29]. Cantu et al. (2020) used bitewing radiographs to assess the performance of a deep learning (DL) network in detecting carious lesions. A total of 3686 radiographs were used, out of which 3293 were used for training while 252 were used as test data. The deep neural network showed higher accuracy compared to dentists and can be used to detect initial caries lesions on bitewing radiographs [30] Park et al. (2021) tested ML prediction models for the detection of early childhood caries compared to traditional regression models. Data of 4195 children (1–5 yrs) were obtained from the Korea National Health and Nutrition Examination survey (2007–2018) and analyzed. ML-based prediction models were able to detect ECC, predict high-risk groups, and suggest treatment, similar to traditional prediction models [31].

## 4. Tooth Fracture

The third most common reason for tooth loss is traumatized or cracked teeth. Early detection and treatment can save a cracked tooth and help retain it. However, cracked teeth often present with discontinuous symptoms, making their detection problematic. Conventional techniques, such as CBCT and intraoral radiographs, have low sensitivity and clarity. Paniagua et al. (2018) developed a novel method capable of detecting, quantifying, and localizing cracked teeth using high-resolution CBCT scans with steerable wavelets and machine learning methods. The performance of ML models was tested using Hr-CBCT scans of healthy teeth with simulated cracks. ML models showed high specificity and sensitivity [32]. Fukuda et al. (2020) used CNN to detect vertical root fractures using 300 panoramic radiographs with 330 vertically fractured teeth with visible fracture lines. Moreover, 80% of the data was used for training while 20% was used as a test data set. Results suggest that CNN can be used as a diagnostic tool for the detection of vertical root fractures [33].

## 5. Periodontal Diseases

Periodontal disease affects more than a billion people globally, destroying alveolar bone and leading to tooth loss. Early diagnosis of periodontal disease using AI can improve the dental status of the patient and improve their overall health and quality of life. Ozden et al. (2015) examined the use of a support vector machine (SVM), decision tree (DT), and ANN to identify and classify periodontal disease. Data from a total of 150 patients were used, 100 as training data and 50 as test data. The three systems classified the data into six types of periodontal conditions. SVM and DT were more accurate as diagnostic support tools compared to ANN [18]. Nakano et al. (2018) used deep learning (DL) to detect oral malodor from microbiota. A total of 90 patients, 45 patients with weak or no malodor, and 45 patients with marked malodor were selected using organoleptic tests and gas chromatography. Gene analysis of the amplified 16s rRNA from the patient’s saliva was carried out. DL was used to classify the samples into malodor and healthy breath. DL showed a predictive accuracy of 97% compared to SVM, which showed 79% [19]. ANN has been used to predict the occurrence of recurrent aphthous ulcers. Gender, serum B12, hemoglobin, serum ferritin, folate levels, candida count in saliva, tooth brushing frequency, the number of fruits and vegetables consumed daily, were related to the occurrence of ulcers [20] Danks et al. (2021) used a deep neural network to measure periodontal bone loss with the help of periapical radiographs. Periapical radiographs of single, double, and triple rooted teeth obtained from 63 patients were used. First, the DNN was trained to detect dental landmarks on the radiographs, and then the periodontal bone loss was measured using these landmarks by the DNN model. The system achieved a total percentage of correct key points of 89.9%. The system showed promising results, which can be further improved upon by experimentation and cross-validation with extended data sets [34] Similarly, a DL model was used to detect and measure periodontal bone loss from panoramic images, which was then used for staging periodontitis. The performance of the DL model was compared to that of three oral radiologists. The staging was done according to the new classification of periodontal and peri-implant disease and conditions [35]. A total of 340 panoramic radiographs were used out of which 90% were used for training while 10% were used for testing. Data augmentation was carried out to increase the data by 64%. The DL model had high accuracy and excellent reliability, suggesting that it can be used for the automatic diagnosis of periodontal disease and as a routine surveillance tool [36].

## 6. Maxillary Sinus Diseases

The maxillary sinuses are structures that are commonly visualized using extraoral radiographs. Automated identification of the sinuses and detection of any pathology in them by AI can lead to a manifold decrease in misdiagnoses. AI can be used as a tool to assist inexperienced dentists. Murata et al. (2018) evaluated the performance of a DL system in diagnosing maxillary sinusitis using panoramic radiographs. The AI performance was compared to that of two radiologists and two residents. The diagnostic performance of the system was similar to that of the radiologists. However, the AI was superior to dental residents [37]. Kim et al. (2019) used radiographs of the maxillary sinus in Water’s view to evaluate the diagnostic performance of the DL system. AI showed a statistically significant improved sensitivity and specificity to radiologists [38]. Mucosal thickening and mucosal retention cysts are often missed by radiologists. Kuwana et al. (2021) used OPG to detect and classify lesions in the maxillary sinus using a DL object detection technique. Detection of the normal maxillary sinus and inflamed maxillary sinus showed 100% sensitivity, whereas the detection sensitivity of mucosal retention cysts was 98% and 89% in the two test data sets that were used. This DL model can be reliably used in a clinical setup [39]. A recent study proposed a CNN model to assist radiologists. The CNN model is capable of detecting and segmenting mucosal thickening and mucosal retention cysts of the maxillary sinus using CBCT images. A total of 890 maxillary sinuses from 445 patients were used in the study. Low dose images were used for training and testing, while full-dose images were used as test data sets. The CNN model performed effectively in both dosage images with no significant difference [40].

## 7. Salivary Gland Diseases

Salivary gland diseases pose a diagnostic challenge to inexperienced dentists due to their confusing and similar morphological resemblances. AI can be a valuable tool in supporting diagnosing diseases of the salivary gland. DL models can, in some instances, be superior to radiologists. In an early Japanese study, researchers used DL to detect fatty degeneration of the salivary gland parenchyma on CT images, which is evident in the case of Sjogren’s syndrome. Of the total 500 CT images, 400 CT images (200 CT images of the control group, 200 CT images of the Sjogren’s syndrome patients) were used as a training dataset while 100 CT images were used as the test data set to analyze the performance of the ML system. The diagnostic performance of DL was equivalent to that of experienced radiologists and significantly superior to inexperienced radiologists [41]. The low incidence and overlapping morphologic features of salivary gland tumors make them challenging to diagnose for clinicians. ML was used to detect malignant salivary gland tumors based on their cytologic appearance. A recursive partitioning algorithm was used to classify 115 malignant tumor samples into 12 morphologic variables. This performance was compared to that of experienced clinicians. The decision tree system test was effective in narrowing down the differential diagnoses, increasing the accuracy of pathological diagnosis [42]. AI has the potential to be used as a tool to predict the recurrence of salivary gland malignancies [43]. Facial nerve injury after surgical treatment for a salivary gland tumor is a severe complication. Chiesa-Estomba et al. (2021) used clinical, radiological, histological, and cytological data to predict the occurrence of facial nerve palsy in patients and reported that AI can be used as an assessment tool for the prediction of facial nerve injury so that both surgeons and patients are well aware of the complications in advance [44].

## 8. Temporomandibular Joint Disorders

Diagnosing TMJ disorders is a challenging issue for inexperienced dentists. ANN systems can simplify and assist in this diagnosis. The performance of an ANN model was tested by recognizing non-reducing disks in patients. The frontal chewing data from 68 patients with normal disks, unilateral and bilateral non-reducing disks, were obtained. Half the data was used to train the ANN system, while the other half was used for testing. The system showed an acceptable level of error and showed potential as a supporting diagnostic tool with an excellent cost/benefit ratio [45]. Bas et al. (2011) conducted a similar study using clinical symptoms. The clinical symptoms and diagnoses of 219 patients were obtained from experienced oral and maxillofacial surgeons. The data from the first 161 patients was used to train the ANN, while the rest of the data was used to test the ANN. The neural network showed acceptable results in diagnosing internal derangements of the TMJ. Additional patient data, clinical data, radiographs, and images could improve the diagnostic capacity of ANN [46]. Iwasaki (2014) applied Bayesian belief network analysis to MRI images to determine the progression of TMJ disorders. A total of 295 cases with 590 sides of TMJs were used with 11 algorithms. The results suggested that the osteoarthritic changes progressed from condyle to articular fossa, and then to the mandibular bone contours. Age, disk form, bony space, and condylar translations were elements that affected disk displacement and bony changes [47]. Choi et al. (2021) developed an AI model to detect osteoarthritis from OPG images. This AI model can be used in clinical setups where a CT facility or a maxillofacial radiologist is not readily available [48]. Orhan et al. (2021) used magnetic resonance images of TMJs to detect TMJ pathologies, such as condylar osseous changes and disk derangements using an AI model [49]. AI models can use a variety of input data to learn. Researchers have even used infrared thermography images of patients with masseter and lateral pterygoid muscles as the area of interest to diagnose TMJ disorders in an AI model [50].

## 9. Osteoporosis

Osteoporosis can be detected on panoramic radiographs. Various indices, such as the gonion index, mental index, mandibular cortical index, and panoramic mandibular index have been used previously to detect osteoporosis [51,52,53,54]. AI could simplify the diagnosis of osteoporosis and buttress the work of radiologists. Kim et al. (2019) evaluated the performance of deep convolutional neural networks (DCNN) based on computer-aided diagnosis (CAD) in diagnosing osteoporosis from panoramic images against radiologists with 10 years of experience. Out of the total 1268 images, 200 images were used as test images. The DCNN- CAD showed results that were highly agreeable with the diagnostic results of the radiologists. DCNN can be used to help dentists in early diagnosis, and referral to specialists [55]. Lee et al. (2020) compared different types of CNN models to assess which model worked best for the diagnosis of osteoporosis and found that the CNN model with transfer learning and fine-tuning was best able to diagnose osteoporosis automatedly [56].

## 10. Oral Cancer and Cervical Lymph Node Metastasis

Oral cancer is the sixth most common malignancy worldwide. Early detection can lead to a better prognosis and a better survival rate [57]. AI can aid in early diagnosis and decrease the mortality and morbidity associated with oral cancer. Nayak et al. (2005) used ANN to discriminate between normal, premalignant, and tissues using laser-induced autofluorescence spectra recordings. This was compared to a principal component analysis of the same issues. The results showed an accuracy of 98.3%, specificity of 100%, and sensitivity of 96.5%, suggesting that this method can have efficient real-time applications [58]. Uthoff et al. (2017) used CNN to detect precancerous and cancerous lesions from autofluorescence images and white light images. CNN was more effective than specialists in diagnosing precancerous and cancerous lesions. The performance of the CNN model can improve with larger data sets [59]. Aubreville et al. (2017) used DL to identify oral cancer based on confocal laser endomicroscopy (CLE) images. This method had an accuracy of 88.3% and a specificity of 90% [60]. Shams et al. (2017) conducted a comparative study to predict the development of oral cancer from oral potentially-malignant lesions using deep neural networks (DNN). DNN was compared to support vector machines, regularized least squares, and multi-layer perception. DNN had a higher accuracy rate of 96% compared to the other systems [61]. These findings were confirmed by Jeyraj et al. (2019). CNN was used to distinguish between cancerous and non-cancerous tissues based on hyperspectral images. Results suggest that CNN can be employed for image-based classification and diagnosis of oral cancer without expert supervision [62]. Recently, a lot of research has taken place in the field of oral cancer research. Many studies have successfully developed AI models that are capable of predicting the occurrence and recurrence of oral cancer [63,64,65,66,67].

Several studies have compared deep learning (DL) systems against experienced radiologists with varied results. Ariji et al. (2014) assessed the performance of DL in the identification of cervical node metastasis using CT images. CT images of 137 positive histologically proven cervical lymph nodes and 314 negative histological lymph nodes from 45 patients with oral squamous cell carcinoma were used. The results of the DL approach were compared against two trained radiologists. The DL network was as accurate as trained radiologists [68]. The researchers also used DL to detect the extra-nodal extension of cervical lymph node metastases. A total of 703 CT images from 51 patients with and without extra-nodal extension were collected and 80% were used as training data while 20% were used as test data. The performance of the DL system was significantly superior to that of the radiologist, suggesting that it can be used as a diagnostic tool for detecting extra-nodal metastasis [69].

Overall, this review on artificial intelligence for diagnosis points towards a positive trend with encouraging results. Neural networks and machine learning appear as effective or better than trained radiologists and clinicians (Table 1) in detecting caries, sinusitis, periodontal disease, and TMJ disorders. Cancer diagnosis by using artificial intelligence models can curate diverse data streams to render judgments, assess risk and referral to specialists (Table 2). Studies on premalignant lesions, lymph nodes, salivary gland tumors, and squamous cell carcinoma show encouraging results for the diagnostic and prognostic value of artificial intelligence. These efforts may reduce mortality rates through early diagnosis and effective therapeutic interventions. These platforms will require large data sets and resources to analyze data to provide a precise and cost-effective diagnosis. In order to be securely integrated into daily clinical procedures, these models are needed to be refined to reach the highest accuracy with specificity and sensitivity. Furthermore, also required are regulatory frameworks for the deployment of these models in clinical practice.

## 11. Prospects and Challenges

AI in dentistry is mostly in the nascent stages. It has yet to enter the realm of day-to-day dentistry. Numerous hurdles remain before it can seamlessly integrate into diagnosis and healthcare. Machine learning requires large volumes of data that are held by private dental setups and institutions. Data sharing and privacy are issues that need to be dealt with through federated guidelines and laws. This can rectify a common drawback reported in most studies: a shortage of data sets. European and American legislative bodies have passed the General Data Protection Act (GDPRA) and the California Consumer Protection Act (CCPA) to limit the risks of data sharing and protect consumer confidentiality [70,71]. Federated data systems similar to VANTAGE6, Personal Health Train (PHT), and DataSHIELD need to be developed so that data can be shared without breaching data security policies [71,72,73]. AI can also convert widely heterogeneous data into curated homogeneous data that is easy to use and interpret. Most of the studies in this review have been supervised image-based studies for the identification of structures or associations. This only provides partial information required for decision-making or treatment. AI capable of unsupervised diagnosis and prediction of diseases needs to be built to reduce subjective errors and provide standardized decisions. A shortage of manpower and resources is emblematic of rural communities. AI-based healthcare initiatives can connect rural and far-flung places with quality health care, benefiting the local population. Prospective randomized control trials and cohort studies have to be performed to evaluate the impact of AI on treatment and to test the outcomes and cost-effectiveness of AI [74,75,76].

## 12. Conclusions

The field of artificial intelligence (AI) is rapidly evolving to fill an ever-expanding niche in medicine and dentistry. Most AI research is still in its nascent stage. Increased availability of patient data can accelerate research into artificial intelligence, machine learning, and neural networks. Today, there are few real-time AI applications integrated into the internal operational process of dental clinics. Research has shown that data-driven AI is reliable, transparent, and in certain cases, better than humans in diagnosis. AI can replicate human functions of reasoning, planning, and problem-solving. Its application can save time and storage, reduce manpower and eliminate human errors in diagnosis. The rise of artificial intelligence in dental care will revolutionize dentistry and usher in wider access to dental health care with better patient outcomes.

## Figures and Tables

**Figure 1 diagnostics-12-01029-f001:**
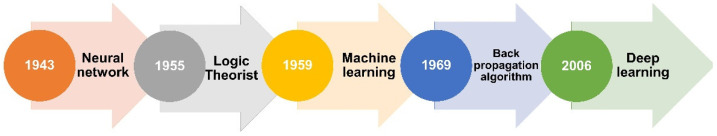
Important milestones in the advancement of AI.

**Figure 2 diagnostics-12-01029-f002:**
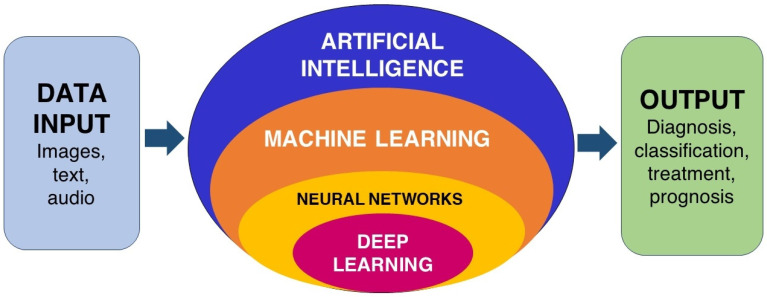
The working of AI in a schematic format.

**Table 1 diagnostics-12-01029-t001:** Summary of studies examining the use of artificial intelligence in dental diagnosis.

Study	Algorithm Used	Study Factor	Modality	Number of Input Data	Performance	Comparison	Outcome
Lee J et al. (2018) [27]	CNN	Dental caries	Periapical radiographs	600	Mean AUC—0.890	4 Dentists	Deep CNN showed a considerably good performance in detecting dental caries in periapical radiographs.
Casalegno et al. (2019) [29]	CNN	Dental caries	Near-infrared transillumination imaging	217	ROC of 83.6% for occlusal caries; ROC of 84.6% for proximal caries	Dentists with clinical experience	CNN showed increased speed and accuracy in detecting dental caries
Cantu et al. (2019) [30]	CNN	Dental caries	Bitewing radiographs	141	Accuracy 0.80; sensitivity 0.75%; specificity 0.83%;	4 experienced dentists	AI model was more accurate than dentists
Radke et al. (2003) [45]	ANN	Disk displacement	Frontal plane jaw recordings from chewing	68	Accuracy 86.8%, specificity 100%, sensitivity 91.8%	None	The proposed model has an acceptable level of error and an excellent cost/benefit ratio.
Park YH et al. (2021) [31]	ML	Early childhood caries	Demographic details, oral hygiene management details, maternal details	4195	AUROC between 0.774 and 0.785	Traditional regression model	Both ML-based and traditional regression models showed favorable performance and can be used as a supporting tool.
Kuwana et al. (2021) [39]	CNN	Maxillary sinus lesions	Panoramic radiographs	1174	Diagnostic accuracy, sensitivity, and specificity were 90–91%, 81–85% and 91–96% for maxillary sinusitis and 97–100%, 80–100% and 100% for maxillary sinus cysts.	None	The proposed deep learning model can be reliably used for detecting the maxillary sinuses and identifying lesions in them.
Murata et al. (2018) [37]	CNN	Maxillary sinusitis	Panoramic radiographs	120	Accuracy 87.5%; sensitivity 86.7%; specificity 88.3%	2 experienced radiologists, 2 dental residents	The AI model can be a supporting tool for inexperienced dentists
Kim et al. (2019) [38]	CNN	Maxillary sinusitis	Water’s view radiographs	200	AUC of 0.93 for temporal; AUC of 0.88 for geographic external	5 radiologists	the AI-based model showed statistically higher performance than radiologists.
Hung KF et al. (2022) [40]	CNN	maxillary sinusitis	Cone-beam computed tomography	890	AUC for detection of mucosal thickening and mucous retention cyst was 0.91 and 0.84 in low dose, and 0.89 and 0.93 for high dose	None	The proposed model can accurately detect mucosal thickening and mucous retention cysts in both low and high-dose protocol CBCT scans.
Danks et al. (2021) [34]	DNN symmetric hourglass architecture	Periodontal bone loss	Periapical radiographs	340	Percentage Correct Keypoints of 83.3% across all root morphologies	Asymmetric hourglass architecture, Resnet	The proposed system showed promising capability in localizing landmarks and periodontal bone loss and performed 1.7% better than the next best architecture.
Chang et al. (2020) [36]	CNN	Periodontal bone loss	Panoramic radiographs	340	Pixel accuracy of 0.93; Jaccard index of 0.92; dice coefficient values of 0.88 for localization of periodontal bone.	None	The proposed model showed high accuracy and excellent reliability in the detection of periodontal bone loss and classification of periodontitis
Ozden et al. (2015) [18]	ANN	Periodontal disease	Risk factors, periodontal data, and radiographic bone loss	150	Performance of SVM & DT was 98%; ANN was 46%	SVM &DT	SVM and DT showed good performance in the classification of periodontal disease while ANN had the worst performance
Devito et al. (2008) [26]	ANN	Proximal caries	Bitewing radiograph	160	ROC curve area of 0.884	25 examiners	ANN could improve the performance of diagnosing proximal caries.
Dar-Odeh et al. (2010) [20]	ANN	Recurrent aphthous ulcers	Predisposing factor and RAU status	96	Accuracy of prediction for network 3 & 8 is 90%; 4,6 & 9 is 80%; 1& 7 is 70%; 2 & 5 is 60%	None	the ANN model seemed to use gender, hematologic and mycologic data, tooth brushing, fruit, and vegetable consumption for the prediction of RAU.
Hung M et al. (2019) [28]	CNN	Root caries	Data set	5135	Accuracy 97.1%; Precision 95.1%; sensitivity 99.6%; specificity 94.3%	Trained medical personnel	Shows good performance and can be clinically implemented.
Iwasaki et al. (2015) [47]	BBN	Temporomandibular disorders	Magnetic resonance imaging	590	Of the 11 BBN algorithms used path conditions using resubstitution validation and 10—fold cross-validation showed an accuracy of >99%	necessary path condition, path condition, greedy search-and-score with Bayesian information criterion, Chow-Liu tree, Rebane-Pearl poly tree, tree augmented naïve Bayes model, maximum log-likelihood, Akaike information criterion, minimum description length, K2 and C4.5	The proposed model can be used to predict the prognosis of TMDs.
Orhan et al. (2021) [49]	ML	Temporomandibular disorders	Magnetic resonance imaging	214	The performance accuracy for condylar changes and disk displacement are 0.77 and 0.74	logistic regression (LR), random forest (RF), decision tree (DT), k-nearest neighbors (KNN), XGBoost, and support vector machine (SVM)	The proposed model using KNN and RF was found to be optimal for predicting TMJ pathologies
Diniz de lima et al. (2021) [50]	ML	Temporomandibular disorders	Infrared thermography	74	Semantic and radiomic-semantic associated ML feature extraction methods and MLP classifier showed statistically good performance in detecting TMDs	KNN, SVM, MLP	ML model associated with infrared thermography can be used for the detection of TMJ pathologies
Bas B et al. (2012) [46]	ANN	TMJ internal derangements	Clinical symptoms and diagnoses	219	Sensitivity and specificity for unilateral and anterior disk displacement with and without reduction were 80% & 95% and 69% & 91%; for bilateral and anterior disk displacement with and without reduction were 37% &100% and 100% & 89% respectively.	Experienced surgeon	The developed model can be used as a supportive diagnostic tool for the diagnoses of subtypes of TMJ internal derangements
Choi et al. (2021) [48]	CNN	TMJ osteoarthritis	Panoramic radiographs	1189	Accuracy of 0.78, the sensitivity of 0.73, and specificity of 0.82	Oral and maxillofacial radiologist	The developed model showed performance equivalent to experts and can be used in general practices where OMFR experts or CT is n
Fukuda et al. (2019) [33]	CNN	Vertical root fracture	Panoramic radiograph	60	The precision of 0.93; Recall of 0.75	2 Radiologists and 1 Endodontist	The CNN model was a promising supportive tool for the detection of vertical root fracture.

**Table 2 diagnostics-12-01029-t002:** Summary of studies examining the use of artificial intelligence in cancer diagnosis.

Study	Algorithm Used	Study Factor	Modality	Number of Input Data	Performance	Comparison	Outcome
Ariji et al. (2019) [69]	CNN	Extra-nodal extension of cervical lymph node	CT images	703	Accuracy of 84%	4 radiologists	The diagnostic performance of the DL model was significantly higher than the radiologists
Lopez—Janeiro et al. (2022) [42]	ML	Malignant salivary gland tumor	Primary tumor resection specimens	115	84–89% of the samples were diagnosed correctly	None	The developed model can be used as a guide for the morphological approach to the diagnosis of malignant salivary gland tumors
Felice et al. (2021) [43]	Decision tree	Malignant salivary gland tumor	Age at diagnosis, gender, salivary gland type, histologic type, surgical margin, tumor stage, node stage, lymphovascular invasion/perineural invasion, type of adjuvant treatment	54	5-year disease-free survival was 62.1%. Important variables to predict recurrence were pathological tumor and node stage. Based on the variables, 3 groups were partitioned as pN0, pT1-2 pN+ and PT3-4 pN+ with 26%, 38% and 75% of recurrence and 73.7%, 57.1% and 34.3% disease-free survival rate, respectively	None	The proposed model can be used to classify patients with salivary gland malignancy and predict the recurrence rate.
Ariji et al. (2019) [68]	CNN	Metastasis of cervical lymph nodes	CT images	441	Accuracy 78.2%; sensitivity 75.4%; specificity 81.1%	not clear	The diagnostic performance of the CNN model is similar to that of radiologists
Nayak et al. (2005) [58]	ANN	Normal, premalignant and malignant conditions	Pulsed laser-induced autofluorescence spectroscopic studies	Not clear	Specificity and sensitivity were 100% and 96.5%	Principal component analysis	ANN showed better performance compared to PCA in the classification of normal, premalignant, and malignant conditions
Shams et al. (2017) [61]	DNN	Oral cancer	Gene expression profiling	86	Accuracy of 96%	support vector machine (SVM), Regularized Least Squares (RLS), multi-layer perceptron (MLP) with backpropagation	The proposed system showed significantly higher performance, which can be easily implemented
Jeyaraj et al. (2019) [62]	CNN	Oral cancer	Hyperspectral images	600	Accuracy of 91.4% for benign tissue and 94.5% for normal tissue	Support vector machine and Deep belief network	The proposed method can be deployed for the automatic classification of
Aubreville et al. (2017) [60]	CNN	oral squamous cell carcinoma	Confocal laser endomicroscopy (CLE) images	7894	AUC 0.96; Mean accuracy sensitivity 86.6%; specificity 90%;	not clear	This method seemed better than the state-of-the-art CLE recognition system
Uthoff et al. (2018) [59]	CNN	Precancerous and cancerous lesions	Autofluorescence and white light imaging	170	sensitivity, specificity, positive, and negative predictive values ranging from 81.25 to 94.94%	None	The proposed model is a low-cost, portable, and easy-to-use system.

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
