# Peer review of "Artificial Intelligence in the Diagnosis of Oral Diseases: Applications and Pitfalls"

_diagnostics, 2022, doi:10.3390/diagnostics12051029_

Round 1

Reviewer 1 Report

In this study, the authors proposed a research trend of AI-enabled diagnosis for oral diseases. The manuscript is well designed and organized, but to improve the quality of the paper some minor revisions will be required. 

  • Please, split Table 1 by each disease or other appropriate criteria.
  • Please, summarize the performance field of Table 1. 

Author Response

Reviewer’s Comments

Reviewer’s Comments

Authors’ reply

Academic editor

Table 1 should be broken into sections.

We are grateful for the time taken by the academic editor in going through our manuscript and providing us with helpful comments. Thank you for your time and patience.

We have made the requested changes and split Table 1 into sections.

Reviewer 1

In this study, the authors proposed a research trend of AI-enabled diagnosis for oral diseases. The manuscript is well designed and organized, but to improve the quality of the paper some minor revisions will be required. 

  • Please, split Table 1 by each disease or other appropriate criteria.

We thank the reviewer for their keen insights. We are grateful to the reviewer for the time spent in assessing our manuscript and providing us invaluable feedback tp improve our manuscript.

We have split Table 1 into two tables summarizing the studies in dental diagnosis (Table 1) and in cancer diagnosis (Table 2)

  • Please, summarize the performance field of Table 1. 

We have added a paragraph summarizing the performance

Line 311-323

Reviewer 2

I have no comments on the review provided to me for the review. The topic is relevant and I hope that in the future there will be more opportunities for use in the widespread practice of artificial intelligence in favor of diagnostics and prevention in dentistry.

We are grateful for the kind words of the reviewer. Thank you for taking time to review our manuscript. We appreciate your time and effort.

Reviewer 2 Report

I have no comments on the review provided to me for the review. The topic is relevant and I hope that in the future there will be more opportunities for use in the widespread practice of artificial intelligence in favor of diagnostics and prevention in dentistry.

Author Response

(The authors gave the same response as above.)
